# Expression of Oncogenic Drivers in 3D Cell Culture Depends on Nuclear ATP Synthesis by NUDT5

**DOI:** 10.3390/cancers11091337

**Published:** 2019-09-10

**Authors:** Katherine E. Pickup, Felicitas Pardow, José Carbonell-Caballero, Antonios Lioutas, José Luis Villanueva-Cañas, Roni H. G. Wright, Miguel Beato

**Affiliations:** 1Gene Regulation, Stem Cells and Cancer Program, Centre for Genomic Regulation (CRG), The Barcelona Institute of Science and Technology (BIST), Dr. Aiguader 88, 08003 Barcelona, Spain; 2Department of Life Science, Universitat Pompeu Fabra (UPF), 08003 Barcelona, Spain

**Keywords:** breast cancer, proliferation, oncospheres, cell–cell communication, cytoskeleton, motility, recurrence, cancer stem cell (CSC), metastasis, 3D cell cultures, EMT

## Abstract

The growth of cancer cells as oncospheres in three-dimensional (3D) culture provides a robust cell model for understanding cancer progression, as well as for early drug discovery and validation. We have previously described a novel pathway in breast cancer cells, whereby ADP (Adenosine diphosphate)-ribose derived from hydrolysis of poly (ADP-Ribose) and pyrophosphate (PPi) are converted to ATP, catalysed by the enzyme NUDT5 (nucleotide diphosphate hydrolase type 5). Overexpression of the *NUDT5* gene in breast and other cancer types is associated with poor prognosis, increased risk of recurrence and metastasis. In order to understand the role of NUDT5 in cancer cell growth, we performed phenotypic and global expression analysis in breast cancer cells grown as oncospheres. Comparison of two-dimensional (2D) versus 3D cancer cell cultures from different tissues of origin suggest that NUDT5 increases the aggressiveness of the disease via the modulation of several key driver genes, including ubiquitin specific peptidase 22 (*USP22*), *RAB35B*, focadhesin (*FOCAD*) and prostagladin E synthase (*PTGES*). NUDT5 functions as a master regulator of key oncogenic pathways and of genes involved in cell adhesion, cancer stem cell (CSC) maintenance and epithelial to mesenchyme transition (EMT). Inhibiting the enzymatic activities of NUDT5 prevents oncosphere formation and precludes the activation of cancer driver genes. These findings highlight NUDT5 as an upstream regulator of tumour drivers and may provide a biomarker for cancer stratification, as well as a novel target for drug discovery for combinatorial drug regimens for the treatment of aggressive cancer types and metastasis.

## 1. Introduction

Metastasis is the main cause of death for breast cancer patients. Metastases are routinely aggressively treated with surgery, followed by chemo- and/or radiotherapy, as few targeted drug regimens are available for metastatic treatment [1]. In order to provide novel targets for drug development, a more complete understanding of the mechanisms, gene expression changes and key regulators involved within the process of metastasis is key. The progression of metastasis can be split into several distinct stages: local proliferation of the primary tumour, local invasion and intravasation (epithelial to mesenchyme transition (EMT)), dissemination and extravasion and finally, colonisation and proliferation within the secondary site. Each stage of the process is dependent on nuclear and gene expression reorganisation to activate or repress specific pathways at each stage [1].

Pluripotent cancer stem cells (CSC) have been identified in the majority of solid and hematologic cancers and have been shown in numerous studies to be the initiators of tumour development, proliferation, metastatic dissemination and colonisation [2]. CSCs are low in number within the primary tumour, however, because of their self-renewal capacities, they have been shown to be resistant both to chemo- and radiotherapy [3,4]. Indeed, several studies have indicated that these kinds of treatments actually promote “stemness”, which may explain why recurrent tumours are often more aggressive than the primary [5]. Stemness involves EMT, whereby the gene expression and phenotypic characteristics of the epithelial cells are changed, increasing the cell capacity for migration, invasion and an avoidance of apoptosis.

We have previously identified NUDIX5 (nucleotide diphosphate linked to moiety-X 5), more generally referred to as NUDT5 (nucleotide diphosphate hydrolase type 5), as a key regulator in hormone receptor positive and serum-starved breast cancer cells exposed to hormone. In response to hormones, PARP1 (poly-ADP-ribose polymerase 1) is activated by CDK2-mediated phosphorylation within the NAD^+^ binding cleft [6] that results in a more open catalytic domain, increasing the activity of the enzyme [7]. Activated PARP1 then rapidly generates poly-ADP-ribose (PAR) from NAD^+^ synthesized within the cell nucleus by NMNAT1 (Nicotinamide Nucleotide Adenylyltransferase 1) within the nucleus of breast cancer cells. The main acceptor for PAR is PARP1 itself, followed by core and linker histones. This massive PARylation gives rise to a more open chromatin structure, aiding the recruitment of chromatin remodelling complexes and the transcription machinery to hormone regulated genes whereby the activated progesterone receptor (PR) binds the following hormone. This massive generation of PAR is transient in nature because of the activity of the counteracting nuclear enzyme, PARG (poly-ADP-ribose glycohydrolase), which cleaves PAR into single ADP-ribose (ADPR) units. This balance of PARP1 and PARG enzymatic activities is essential to ensure that the metabolic substrates required for PAR synthesis, namely ATP and NAD^+^, are not exhausted to prevent cell death by parthanatos [8]. The importance of parthanatos avoidance and the consequently tight regulated balance of PARG and PARP1 enzymatic activities is found in many other biological situations in mammalian cells, not only in hormone induction, but also including DNA damage [9], replication [10] and cell cycle, chromatin remodelling and gene expression [11].

NUDT5 is a homodimer (Protein Databank; PDB ID:2DSC [12]) that was identified by mass spectroscopy as an interactor of PAR following hormones [13]. Subsequently, we found that NUDT5 not only hydrolyses ADP-ribose (ADPR) to AMP and ribose-5-phosphate (R5P), but in the presence of pyrophosphate (PPi) can also act as a pyrophosphorylase, generating ATP and R5P [13]. This later reaction requires the dephosphorylation of NUDT5 at T45, destabilizing the homodimer and opening the substrate binding groove to enable PPi entry [13].

*NUDT5* is overexpressed in breast cancer compared to normal tissue and stratifying patients on the basis that elevated expression levels of *NUDT5* predicts a poorer prognosis for patients [13]. However, the mechanism by which NUDT5 drives cancer progression and is exploited by cancer cells is not known. In this article, we focus on identifying NUDT5-dependent pathways in cancer progression and metastasis. To this end, we used as an experimental model the generation and maintenance of oncospheres starting from breast cancer cells. These structures can be generated with multiple cancer cell lines and are known to be enriched in CSCs, on the basis of the expression of CSC genes, including *ALDH1*, *SOX2*, *CD44+/CD24−* [14,15]. They are also enriched in markers of EMT, including *CTNNB1*, *EGR1*, *ERBB2* and *MUC1* [16,17]. Thus, compared to cells in two-dimensional (2D) culture conditions, oncospheres provide a more realistic model for understanding cancer progression in vivo [18], and are appropriate for performing early stage cancer drug discovery validation and optimisation [19]. The data presented here may aid in the development of more targeted therapies for cancer patients with advanced disease.

## 2. Results and Discussion

### 2.1. NUDT5 in Cancer

#### 2.1.1. NUDT5 in Expression in Human Cancers Correlates with an Aggressive Phenotype

Analysis of the TGCA data revealed that NUDT5 mRNA levels were elevated in tumours compared to normal tissue across the spectrum of cancer types (Appendix A, abbreviations of cancer types; Appendix A). A positive staining of NUDT5 was also found within several cancer types of different origin; indeed over 40% of liver and 30% of breast cancers show a positive staining with an antibody specific for NUDT5 (Appendix A). Immuno-histological staining of NUDT5 in patient tumour samples and normal samples showed a clear increase in tumour versus normal tissues (Appendix A), with metadata for the samples given in Appendix A. Complementary to the observation that NUDT5 can be used as a prognostic marker in breast cancer [13] elevated NUDT5 levels were also prognostic in kidney, adrenal, brain and liver patient datasets (Appendix A). Moreover, breast cancer patients with increased expression of NUDT5 showed an enhanced at risk of recurrence and metastasis (Appendix A). Collectively, these findings led us to the hypothesis that elevated levels of NUDT5 were prognostic of a poor outcome due to the development of a more aggressive cancer disease phenotype.

#### 2.1.2. Effect of Nudt5 Knockdown on Cell Growth in 2D Cultures, Cell Migration and Colony Formation Assays

To ascertain whether NUDT5 is indeed driving a more aggressive cancer phenotype, we generated T47D breast cancer cells stably depleted of NUDT5, as well as the corresponding rescue cell lines. First, we used a Tet-Off system and specific shRNAs to generate stable cell lines, wherein the endogenous NUDT5 was over 95% depleted (NUDT5^KD^) (Figure 1A–C and Appendix A). Using NUDT5^KD^ cells and a pKAR plasmid [20] containing the Tet-Off driving, an shRNA resistant FLAG-tagged NUDT5 mutant rescue cell line was generated (NUDT5^RES^). An efficient rescue was reproducibly observed across several stable clones at both the protein and mRNA levels (Figure 1B–D).

Surprisingly, breast cancer cell proliferation was not significantly affected in NUDT5^KD^ cell lines when grown in 2D culture (Figure 1E). However, in wound healing and clonogenic assays, we observed a significant defect in the NUDT5^KD^ cell lines (Figure 1F–I), suggesting a role for NUDT5 in cancer cell motility, cell migration or on growth in low cell density conditions.

#### 2.1.3. Characterisation of the Expression Profile Changes in T47D Oncosphere Model

These observations led us to consider that NUDT5 had no effect on the cell cycle under normal growth conditions but could drive changes in the cancer cell phenotype, perhaps promoting an EMT and/or stem cell-like state. One experimental model used often for understanding these processes is growing the cancer cell lines in non-adherent conditions as three-dimensional (3D) oncospheres. During this process, the majority of seeded single cells die during the initial days of culture because of the challenge of growing in non-adherent surfaces [14], and the surviving cells express markers of EMT and stemness, and initiate the formation of spheres within 5–6 days [2,15].

Therefore, we generated oncospheres in T47D NUDT5^RES^ cell lines and, in order to gain a more global understanding of the gene expression changes which took place following the transition between 2D and 3D (Figure 2A), RNA-seq was carried out in NUDT5^RES^ grown in the two conditions (Appendix A, Appendix A). A significant portion of the 5850 genes, which significantly changed their expression between 2D and 3D culture conditions (Figure 2B), were found within a curated set of 480 oncosphere-regulated genes (Figure 2C and Appendix A), including genes involved in EMT, oncosphere growth, signalling, adhesion, pluripotency and stem cells [21,22,23] referring to the dEMT database (http://dbemt.bioinfo-minzhao.org). The most significant functional enrichment and expression changes between 2D and 3D were observed for genes involved in EMT (108 genes) (Figure 2C–D). Upregulation of several membrane proteins: EMP3, FGFR1, GHR and IGF2R, involved specifically in mammosphere growth [22] occurred, and canonical stem cell markers EPCAM and SOX2 [23] were also up-regulated during the transition from 2D to 3D (Figure 2E).

Tumour markers, including those within the carcinoembryonic antigen (CEA)-related cell adhesion molecules (CEACAM) family, have mainly been used for the diagnosis and monitoring of colon carcinoma recurrence following surgery [24], although elevated levels have also shown clinical use in cancer monitorization for several cancers, including lung, pancreatic, liver, cervical and breast cancer [25,26,27,28,29,30]. Elevated levels of CEACAM5 were found in breast cancers and were identified as a metastatic driver [31]. We found significantly elevated levels of CEACAM 5, 6 and 7 in T47D cells grown in 3D cell culture compared with 2D culture (Figure 2F). In addition, the transmembrane protein MUC1 is often overexpressed in metastatic cancers and—like CEACAM—is used to monitor metastatic progression [32,33]. MUC1 is known to upregulate the expression of EMT drivers and to reduce the contacts between cancer cells, facilitating basement membrane invasion. We found that the transmembrane proteins MUC1, MUCL1 and MUC5 showed a significant increase in expression in oncospheres compared with cells in 2D culture (Figure 2F). These findings further strengthen the oncosphere model as an appropriate model for breast cancer studies.

Furthermore, changes within the Mucin1 complex (cSRC-MUC1) were also found in comparing cells grown in 2D and 3D cultures. MUC1 co-localisation with SRC1 on the plasma membrane has been shown to be essential for SRC1 signalling in tumour development [34]. These alterations were also confirmed following CORUM (comprehensive resource of mammalian protein complexes) analysis of the changes in gene expression (Figure 2G). In addition, we found changes in Caveolin1-VDAC-ESR1 complexes, which are involved in cell–cell communication, membrane mobilisation and cell adhesion (Figure 2G), and in complexes regulating transcription, chromosome organisation and nuclear architecture from 2D to 3D, including the cohesin complex, SIN3-ING1b complex 1 and II, ING-p300-PCNA and EGR-EP300 complexes (Figure 2G and Appendix A), in addition to alterations effecting the stem cell transcription factor complexes OCT1-SOX2, PAX6-SOX2 and OCT4-SOX2. Gene ontology and pathway enrichment showed a significant alteration in key regulatory processes, including mitosis, cell cycle, DNA repair, and chromatin cohesion (Appendix A and Appendix A). Analysis of the transcription binding site motifs enriched within promoters (1.5 kbp up- and downstream of the TSS (Transcription start site) of genes involved in the transition between 2D and 3D showed a significant enrichment in SP1, ZNF281, ZNF740, MZF1 and RREB1 motifs, wherein not only were the binding sites enriched, but the expression of the transcription factors was also increased at the RNA level from 2D to 3D (Figure 2H and Appendix A), suggesting a system-wide reorganisation of the expression program. Notably, SP1 and ZNF281 were predictive of a poor outcome on the basis of our analysis of publically available patient datasets (Appendix A) and previous observations [30,35]; MZF1 is a known oncogenic transcription factor involved in tumour progression [36,37]. Overall the overlap with previously published datasets, as well as the enrichment in functions associated with a more stem-like phenotype and clinical tumour markers during the transition from 2D to 3D, validates the oncosphere model as a representation of a more CSC-like phenotype.

#### 2.1.4. NUDT5 Knock-Down Prevents Oncosphere Formation

Unlike NUDT5^RES^ cells, NUDT5^KD^ cells cannot form oncospheres (Figure 3A). Indeed the “structures” that remain after 5 days are smaller, less abundant and lose their spherical appearance (Figure 3A–C). NUDT^RES^ oncospheres show no phenotypic changes compared to T47D wild type cells (Appendix A). As expected, NUDT5^RES^ oncospheres, but not NUDT5^KD^, were enriched in cells expressing cancer stem cell (CSC) markers, as exemplified by the *CD44+/CD24−* ratio that was also low in cells growing in 2D cultures (Figure 3D), the alcohol dehydrogenase levels (*ALDH1*, Figure 3E) and the cell adhesion proteins including caveolin (*CAV1*, Appendix A). Moreover, the self-renewal pluripotent capacity of the oncospheres derived from NUDT5^RES^ cells was significantly higher than those for NUDT5^KD^ cells (Figure 3F). These findings suggest that the enriched CSC phenotype within oncospheres was dependent on the expression of *NUDT5*.

#### 2.1.5. NUDT5 Promotes the Expression of Genes Involved in EMT and in the Aggressive Tumour Phenotype

In order to understand the role of NUDT5 in the characteristic changes in gene expression observed in T47D cells grown in 3D culture conditions, we carried out RNA-seq analysis in NUDT5^RES^ versus 3D NUDT5^KD^ grown in 3D conditions. Neither NUDT5 mRNA nor protein levels changed during in 3D conditions relative to 2D conditions (Figure 4A and Appendix A). However, a comparison of NUDT5^RES^ versus NUDT5^KD^ both in 3D conditions showed a differential expression of 128 genes (Figure 4B,C). Comparing the genes dependent on NUDT5 at both 2D (Appendix A) and 3D cell culture conditions revealed a core of 60 genes (including NUDT5), the expression of which was dependent on NUDT5, independent of the culture conditions (Figure 4C). Closer inspection of this set of NUDT5-dependent genes revealed that expression was regulated in the same direction, and to a similar extend independent of the 2D or 3D culture condition (Figure 4D). On the basis of these findings, we considered these set of NUDT5-dependent genes to explain the inability of cancer cells to form oncospheres in the absence of NUDT5.

The comparison of RNA-seq from NUDT5^KD^ and NUDT5^RES^ in 3D culture conditions also revealed 68 gene, which induced only in 3D culture conditions and not in 2D conditions (Figure 4C). These set of genes were more likely candidates for explaining the need of NUDT5 for oncosphere formation. Disease terms enriched in this set of 68 genes included breast and, specifically, breast neoplasms (Figure 4C). Analysis of reactome pathways indicated that this set of genes were enriched in extracellular matrix organisation, Notch and Hedgehog signalling pathways (Figure 4E). The Notch signalling pathway has been shown to be involved in invasive breast cancer and key proteins within the pathway identified as targets for cancer therapy drug discovery [38], and mammosphere growth is inhibited following inhibition of Notch signalling [39]. The Hedgehog signalling pathway is involved in cell survival, cell self-renewal and CSC [40], and its aberrant expression has been observed in over 25% of cancers, including breast cancer [41]. Indeed, it has been shown for both Notch and Hedgehog that the CSC niche in mammospheres depends upon signalling via these pathways [42,43].

#### 2.1.6. NUDT5-Catalysed Synthesis of Nuclear ATP is Required for Oncosphere Formation

Enzymatically, NUDT5 has the ability to generate AMP or ATP depending on its phosphorylation status and the availability of substrates PPi [44]. Given that the gene expression data demonstrates that NUDT5 is essential for oncosphere generation, and for enrichment in gene expression signatures for CSC and tumour markers, we next used NUDT5 and to the need for NUDT5 enzymatic activity for oncosphere generation. Page and colleagues characterised the NUDT5 specific inhibitor TH5427 [45], which has been shown to block both the AMP and ATP generating activities of NUDT5. In the presence of TH5427, the growth of NUDT5^RES^ oncospheres was inhibited, showing that not only the presence of NUDT5 but also its enzymatic activity is required for oncosphere generation (Figure 5A–C).

As TH5427 inhibits both the AMP- and ATP-generating activities of NUDT5, and no specific inhibitor targeting only the ATP-generating activities of NUDT5 was available, we took advantage of the Tet-Off cassette present within the NUDT5^RES^ cells (Figure 5D). Treatment of these cells with doxycycline for 48 h induced the depletion of the FLAG-NUDT5 rescue mRNA and total protein levels (Figure 5E). NUDT5^RES^ cells were grown as oncospheres for 2 days, prior to the doxycycline treatment for an additional 3 days, similar to the NUDT5^KD^ cell lines (Figure 3A). This treatment resulted in phenotypically abnormal, significantly fewer and smaller oncospheres in general (Figure 5F,G and Appendix A). The effect of NUDT5 inhibition on oncosphere growth was not limited to cells of breast origin but was observed in multiple cancer cell lines (BT474, HEK293T, U2OS, MNNG-HOS) grown as oncospheres (Appendix A).

We have previously shown that hormonal induction leads to NUDT5 de-phosphorylation at T45 that results in a catalytic shift in NUDT5 conformation, favouring the ATP-generating enzyme [13]. We hypothesised that in 3D culture conditions, the activity of NUDT5 could also change in a similar way. To explore this possibility, we performed immunofluorescent staining and western blot analysis using a phospho-T45 NUDT5-specific antibody in NUDT5^RES^ cell lines grown in 2D and 3D conditions. We found that NUDT5 was phosphorylated at T45 in 2D culture conditions but not in 3D conditions (Figure 5H). To test the functional significance of this finding and given the lack of a specific ATP inhibitor, we generated stable cell lines, whereby endogenous NUDT5 was knocked down using shRNA, and rescued with an sh-resistant NUDT5 containing the T45D phosphomimetic mutation, a mutation which we have previously shown to be able to generate AMP but is defective in ATP generation, both in vitro and in vivo [13]. The inability of these cells to form oncospheres (Figure 5I and Appendix A) confirmed the idea that it is the ATP-generating ability that is required for a successful oncosphere generation and maintenance in 3D cultures of breast cancer cells. Moreover, the significant reduction in the self-renewal capacity of these cells in the absence or inhibition of either AMP and ATP, or ATP-only generating activities of NUDT5 indicates a loss of CSC and EMT gene expression changes that are required to maintain growth as oncospheres (Figure 5J), which was confirmed by qRT-PCR for ALDH1, EGFR and MUCIN1 following either knockdown, inhibition or the use of the NUDT5 phosphomimetic mutant (Figure 5K).

#### 2.1.7. NUDT5-Dependent Genes are Required for Oncosphere Growth and Correlate with Poor Cancer Prognosis

The oncosphere expression data (Figure 4C) not only provided insight into known CSC, EMT or tumour markers, but also identified novel NUDT5-dependent genes relevant for oncosphere growth. Therefore, we focused our attention on several of these genes based on the strong activation of their expression in the oncosphere, which was dependent on NUDT5 (Figure 6A). Prostagladin E synthase (PTGES) is known to be associated with an increased risk of breast cancer [46]; focadhesin (FOCAD) is a transmembrane protein of which very little is known, except a putative tumour suppressor role in gliomas [47]; the small GTPase RAB35B has putative oncogenic properties [48] and its inhibition reduces exosome release, tumour growth, CSC and metastasis [49]; the ubiquitin specific peptidase 22 (USP22) has been identified as 1 of 11 known “death from cancer” signature genes [50], promoting lethal tumour progression, being associated with end stage disease and CSC [51,52] and has gained interest as a cancer therapy target [53]. We validated the significant reduction in expression of some of these genes in oncospheres, following the inhibition of NUDT5 by qRT-PCR (Figure 6B). The co-expression correlation of NUDT5 with FOCAD, PTGES, USP22 and RAB35 was further explored using TGCA patient datasets in breast cancer and showed a significant correlation in all cases (Figure 6C). As these genes are regulated by NUDT5 and their expression patterns correlate in patients, we predicted that stratifying patients on the basis of the expression levels of the most NUDT5-dependent genes (FOCAD and USP22) would also predict a poorer patient outcome similar to that observed for NUDT5. This prediction was clearly confirmed in patient datasets (Figure 6D,E).

It remains to be established how the 3D culture conditions lead to de-phosphorylation of NUDT5 and in what form the nuclear synthesis of ATP favours the expression of these set of tumour driver genes. Overall these results support that tumours with elevated expression of these NUDT5-dependent genes are associated with poor prognosis, likely to be explained by the more aggressive cancer phenotype providing novel targets for combinatorial strategies.

## 3. Materials and Methods 

### 3.1. 2D and 3D Cell Culture

Transfections were performed using Lipofectamine^2000^ (Invitrogen, Carlsbad, California, USA) according to the manufacturer’s instructions. For oncosphere generation, cells were seeded at 10,000 cells/well in 2 mL of DMEM-F12 (supplemented with 20 ng/mL human EGF and B27) media in lo-bind ultra-low attachment plates. Plates were left un-agitated for 5–6 days and the mammospheres were counted and photographed on an Evos microscope. Mammosphere area was measured using Fiji.

#### 3.1.1. Stable Cell Line Generation

NUDT5^RES^ and NUDT5^KD^ stable cell lines were generated using the pKAR system [20], as described. Briefly, shRNA targeting endogenous *NUDT5* was cloned into pKAR using Bbc1 and Xba1, generating a pKAR-*shNUDT5* plasmid (used for generation of NUDT5^KD^ cell lines). Subsequently, the coding sequence of an sh-resistant *NUDT5* was either wild type NUDT5^RES^, or T45D NUDT5^T45D^ was generated by PCR out of a myc-NUDT5 vector, flanked with Apa1 and EcoR1. Silent mutations to introduce resistance to the shRNA were introduced with site-directed mutagenesis, and the sequences were cloned in frame upstream of FLAG within pKAR-*shNUDT5* to generate and induce knockdown sh-resistant contrast for the generation of NUDT5 rescue cell lines (NUDT5^RES^ and NUDT5^45D^). Individual stable cell clones were selected and expanded following selection with G418 (100 μg /mL).

#### 3.1.2. Secondary Oncosphere Assays

The primary mammospheres grown for 5 days were trypsinised for 30 min prior to manual dissociation into a single cell suspension (25-gauge needle). Single cells were re-plated in low-well, low-adherence plates using 1 cell per well. Oncosphere self-renewal was calculated by dividing the number of spheres counted by the number of total cell-containing seeded wells.

#### 3.1.3. RNA Extraction and RT-PCR

Total RNA and reverse transcription was carried out according to the manufacturer’s instructions (RNAeasy and Superscript II Invitrogen, respectively). Complementary DNA was quantified by qPCR using Roche Lightcycler capillaries (Roche, Basel, Switzerland), as previously described [6]. For each gene product, relative RNA abundance was calculated using the standard curve method and expressed as relative RNA abundance after normalizing against the human *GAPDH*. Data presented represent the average ± SEM for at least three independent biological replicates. Primers used for RT-qPCR are available upon request.

#### 3.1.4. Protein Extract Preparation, Western Blotting

2D or 3D cell cultures were similarly prepared for protein analysis. Briefly, cells were lysed (1% NP40, 150 mM NaCl, 50 mM Tris-HCl) and following total protein quantification, equal quantities of cell extract were separated by SDS-PAGE western blot gels and the protein of interest was analysed by western blotting using PARP1- (Cell Signalling technologies, Danvers, Massachusetts, USA) and NUDT5 (Abcam, Cambridge, UK)-specific antibodies.

#### 3.1.5. Immunofluorescence, Image Acquisition and Analysis

Two-dimensional or three-dimensional NUDT5 knockdown or knockdown and rescue cultures were fixed in a similar manner with several differences. Two-dimensional cells were grown on round 10 mm glass coverslips, prior to fixation with 4% paraformaldehyde in PBS for 5 min, as compared to 3D cultures, which were grown in glass-bottom cell chambers (Lab Tek II, Sigma Aldrich, Sankt Gallen, Switzerland). Following fixation, both types of culture were permeabilized with PBS 0.1% Triton X-100 (PBS-T) at room temperature for 5 min. Fixed cells were then blocked with 5% BSA and 0.1% Triton X-100 in PBS for 20 min at room temperature and incubated overnight with primary antibodies for phosphor-T45 NUDT5 (generated in-house). Following three washes with PBS-T, samples were incubated with secondary antibodies at a dilution of 1:2000 (AlexaFluor 594 anti-rabbit, Invitrogen-Molecular Probes, Eugene, California, USA) for 1 h at room temperature, followed by three additional washes with PBS-T. Images of the mounted samples were then acquired with a Leica SP5 confocal microscope using an HC PL 40×/1.4 oil immersion lens using the Leica acquisition software.

#### 3.1.6. BrdU Cell Proliferation

Cell proliferation was assessed using chemiluminescent BrdU assay (Roche, Basel, Switzerland) according to the manufacturer’s instructions. Experiments were carried out in triplicate and presented as mean ± SEM.

#### 3.1.7. Scratch Wound Healing Assays

For migration assays, cells were plated out in 24-well plates at a seeding density to ensure confluence within twenty-four hours. Twenty-four hours later, a scratch was made using a 200 μL pipette tip. Cell debris was removed by washing, and the images of the gap were taken for the following 3−4 days. Data are represented as relative gap width of three independent experiments carried out in duplicate.

#### 3.1.8. Colony Formation Assays

NUDT5^RES^ and NUDT5^KD^ cells were plated out in 10 cm plates at 1000 cells per plate. Medium was replaced over a 10-days period, allowing colonies to form. Surviving colonies were stained using 5% crystal violet solution (10% methanol).

### 3.2. Bioinformatic Procedures

#### 3.2.1. RNA-Seq Data Processing

The initial samples are available in Appendix A. Quality control was evaluated using FastQC [54], and adapter sequences were removed using Trimmomatic [55] with the parameter values: PE, ILLUMINACLIP: TruSeq3-PE-2.fa:2:30:12:1:true LEADING:3 TRAILING:3 MAXINFO:50:0.999 MINLEN:36.

Transcript-level quantification was performed with Kallisto (v. 0.43.0) [56] against the Hg38 version of the genome using −b = 100. We discarded two samples (rw_043_01_02_rnaseq and rw_044_01_02_rnaseq) that appeared as outliers in the PCA (Appendix A) after checking that indeed their distribution of number of counts was much lower (Appendix A).

#### 3.2.2. Gene Expression Analysis

Differential expression analysis was performed using the Sleuth package [57] (v. 0.30.0). The DE analysis between 2D and 3D was done by aggregating transcript *p*-values into genes using the Lancaster method [58]. We used the likelihood ratio test (lrt) using two different reduced: full models to test for the space and condition effects, respectively, while correcting for the sequencing batch effect. For this test, Sleuth does not return a fold change or equivalent metric; therefore, we calculated a log2 fold change using the averages across conditions and a pseudocount of 0.01 log2(tpm_wt + 0.01/tpm_kd + 0.01).

The comparisons within spaces (2D_wt vs. 2D_kd and 3D_wt vs. 3D_kd) were done using the Wald test at the transcript level. We used a single variable with four values (2D_wt, 2D_kd, 3D_wt, 3D_kd) and generated contrasts to test for specific comparisons while correcting for the sequencing batch effect.

Enriched gene expression signatures associated with EMT and the corresponding global prioritization scores shown in Figure 2D were obtained from dbEMT online resource [59].

#### 3.2.3. Gene Ontology Analysis

GO biological process (BP), molecular function (MF), reactome pathway and CORUM complex analysis was performed using the online analysis database, Enrichr [60]. Enriched medical subject headings (MeSH) were calculated using the SEEK Princeton database.

#### 3.2.4. Kaplan Meyer Survival Analysis

Kaplan Meyer stratifying of patients based on gene expression levels of NUDT5 and USP22 was performed using the PrognoScan database, GEPIA (Gene Expression Profiling Interactive Analysis) [61] and Human protein Atlas [62]. NUDT5 overexpression was considered significant containing a corrected *p*-value of less than 0.05, the number of patients considered in each cohort tested is given on the graphs.

## 4. Conclusions

The work presented here shed light onto the mechanism by which elevated levels of NUDT5 observed in cancer patients is predictive of a poor outcome and prognosis (Appendix A). We have shown that elevated levels of NUDT5 and, specifically, ATP enzymatic activity are essential in driving the expression of known CSC and EMT genes, as well as novel tumour drivers detected using the 3D oncosphere model (Appendix A). The NUDT5-dependent expression of known EMT and CSC genes and of new cancer driver genes (*FOCAD* and *USP22*), which also predict a poorer overall outcome in patient datasets, strongly indicate that any future drug discovery focusing on an ATP-specific inhibitor of NUDT5 would be potentially beneficial for the treatment of cancers, including breast and ovarian cancer; in reducing side effects; improving patient health and lifting economic burden.

## Figures and Tables

**Figure 1 cancers-11-01337-f001:**
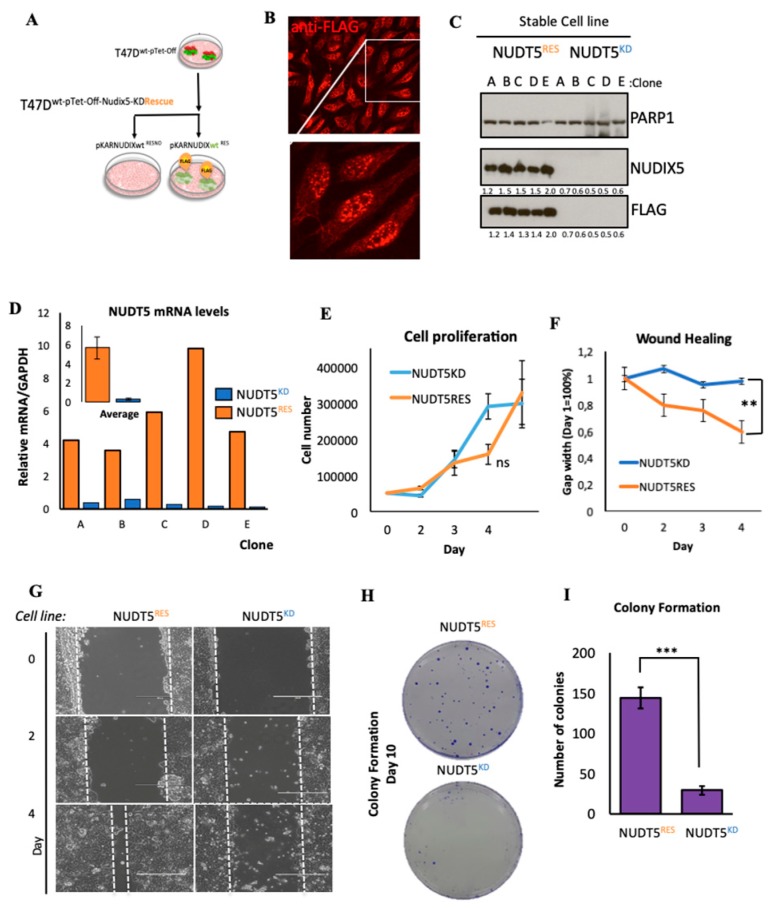
Characterisation of NUDT5^KD^ and NUDT5^RES^ cell lines in 2D cell culture. (**A**) Experimental approach for the stable knockdown and inducible pKAR rescue of NUDT5 (nucleotide diphosphate hydrolase type 5) in T47D^M^ cells. Protein expression of FLAG-NUDT5 using anti-FLAG antibodies in T47D^M^ cells depleted of NUDT5 (NUDT5^KD^) or rescued with the stable expression of the wild type FLAG-NUDT5 (NUDT5^RES^) visualised by immunofluorescence (**B**) and by western blotting (**C**) (PARP1 (poly-ADP-ribose polymerase 1) is shown as a loading control) in several clonal cell lines. Intensity ratio of the observed bands NUDT5 and FLAG are standardised against PARP1 and the values shown below. The full western blot is shown in Appendix A. (**D**) Relative mRNA expression (normalised against GAPDH) of *NUDT5* in NUDT5^KD^ and NUDT5^RES^ cell lines for each clonal cell line. The inset shows the average of all clones. (**E**) Cell proliferation assays measured by BrdU incorporation in NUDT5^KD^ and NUDT5^RES^ cell lines. (**F**) Quantification of cell migration assays in NUDT5^KD^ versus NUDT5^RES^ cell lines. (**G**) Representative images of cell migration assays at different time points, NUDT5^KD^ versus NUDT5^RES^ cell lines. (**H**) Colony formation assay with NUDT5^KD^ versus NUDT5^RES^ cell lines after 10 days in culture. (**I**) Quantification of data shown in (H), showing a statistically significant decrease (*p* < 0.01) in surviving colonies in the absence of NUDT5. The mean ± SEM (standard error of the mean) of three replicas are shown.

**Figure 2 cancers-11-01337-f002:**
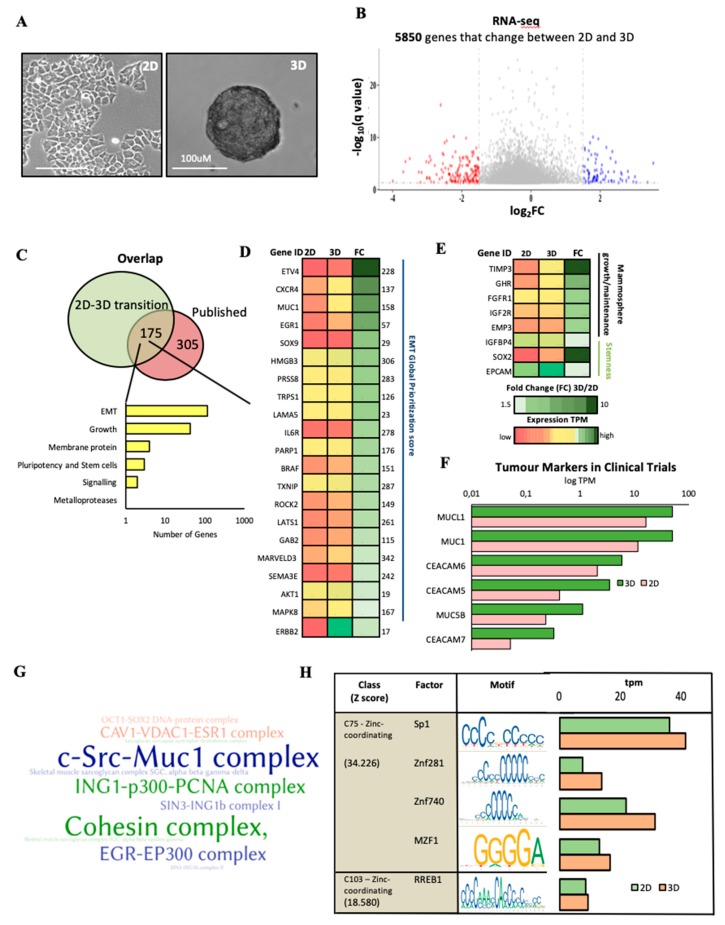
Analysis of the gene expression changes occurring during a transition from 2D to three-dimensional (3D) growth conditions. (**A**) Representative images of NUDT5^RES^ cell lines grown in 2D and 3D culture conditions, scale bar = 100 μM. (**B**) RNA-seq differential expression analysis of NUDT5^RES^ cells grown in 3D and in 2D culture conditions. Changes in gene expression were considered significant if the log fold change was higher than 1.5 (fold change (FC) > 1.5) and the *q*-value was lower than 0.01 (*q* < 0.01). (**C**) Venn diagram showing the overlap of genes expressed differentially in 3D culture and a manually curated list of oncosphere expression datasets (Appendix A). A histogram of the more enriched functional groups in the overlapping cluster of 175 genes is shown at the right margin. (**D**) Heat map presentation of the expression levels in genes involved in epithelial to mesenchyme transition (EMT) in cells growing in 2D and 3D conditions, ranked following the fold change (FC). The EMT Global Prioritization Score is indicated in the right margin. (**E**) Heat map representation of the RNA expression levels (2D and 3D) of genes involved in stemness and mammosphere growth, ranked in descending order of 3D/2D fold change (FC). (**F**) Expression levels of the tumour marker families carcinoembryonic antigen (CEA)-related cell adhesion molecules (CEACAM) and MUCIN in 2D and 3D RNA sequence samples. (**G**) Word cloud showing CORUM (comprehensive resource of mammalian protein complexes) database analysis detailing the significantly altered protein complexes in the genes altered in 2D to 3D transition. (**H**) Transcription factor binding site motifs (central column) enriched within promoters of genes expressed in 3D. The histogram on the right shows the expression levels of the corresponding transcription factors in cells growing in 2D (green) and 3D (orange).

**Figure 3 cancers-11-01337-f003:**
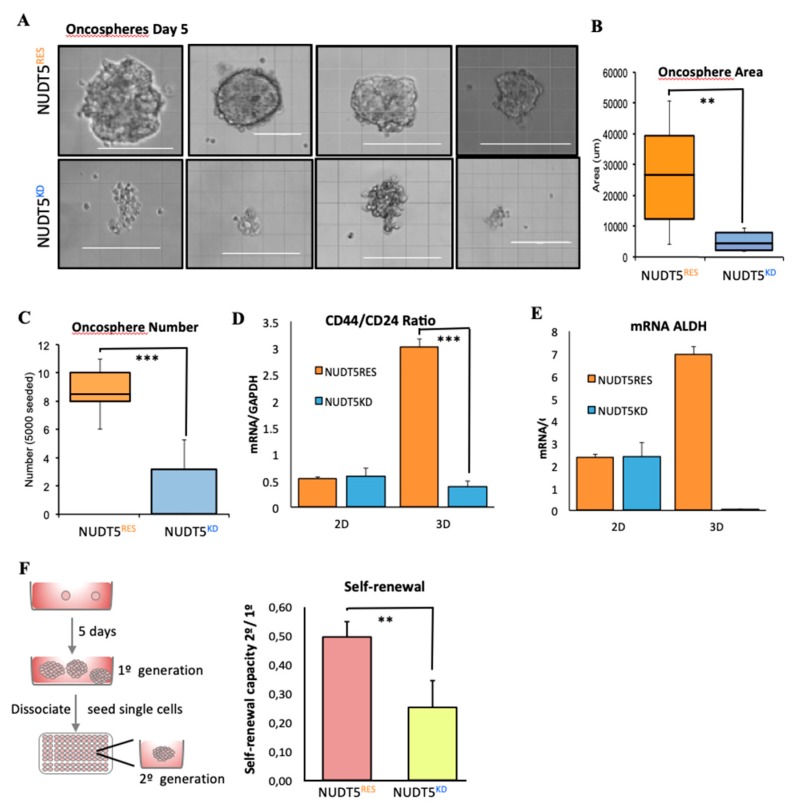
NUDT5 is essential for oncosphere growth and maintenance. (**A**) Bright field images of oncospheres formed following 5 days in culture in NUDT5^KD^ or NUDT5^RES^ cell lines. Scale bar represents 200 μM. (**B**) Quantification of the area of oncospheres formed by NUDT5^KD^ or NUDT5^RES^ cell lines mean ± SEM. (**C**) Quantification of the number of oncospheres formed in NUDT5^KD^ or NUDT5^RES^ cell lines, mean ± SEM. (**D**) CD44 and CD24 ratio in NUDT5^KD^ or NUDT5^RES^ cell lines, grown in 2D and 3D; mean ± SEM. (**E**) *ALDH1* mRNA levels ratio in NUDT5^KD^ or NUDT5^RES^ cell lines, grown in 2D and 3D; mean ± SEM. (**F**) Self-renewal capacity of second generation oncospheres in NUDT5^KD^ or NUDT5^RES^ cell lines; mean ± SEM.

**Figure 4 cancers-11-01337-f004:**
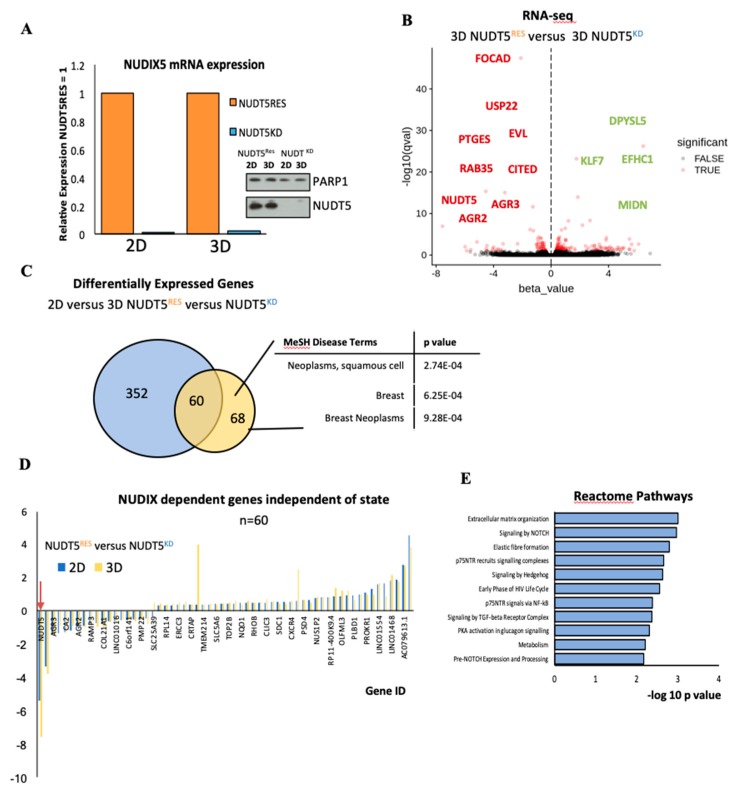
Gene expression changes induced specifically in 3D culture conditions depend on NUDT5. (**A**) Relative mRNA expression (normalised against *GAPDH*) of *NUDT5* in NUDT5^KD^ and NUDT5^RES^ cell lines in 2D and 3D conditions, inset protein expression of NUDT5. Full western blot is found in Appendix A. (**B**) Volcano plot from RNA-seq differential expression analysis comparing 3D-NUDT5^KD^ and 3D-NUDT5^RES^ cell lines. (**C**) Venn diagram showing the overlap between 2D and 3D analysis. Genes changed in 2D dependent on NUDT5 shown in blue (2D-NUDT5^KD^ versus 2D-NUDT5^RES^) and genes changed in 3D dependent on NUDT5 in yellow (3D-NUDT5^KD^ versus 3D-NUDT5^RES^). Enriched medical disease terms (MeSH) within the 3D specific data set are shown (right panel). (**D**) mRNA expression of 60 genes, which were dependent on NUDT5 at both the 2D and 3D states; NUDT5 is indicated by an arrow. (**E**) Reactome pathway analysis showing the significantly (*p* < 0.05) enriched pathways which were dependent on NUDT5 in 3D.

**Figure 5 cancers-11-01337-f005:**
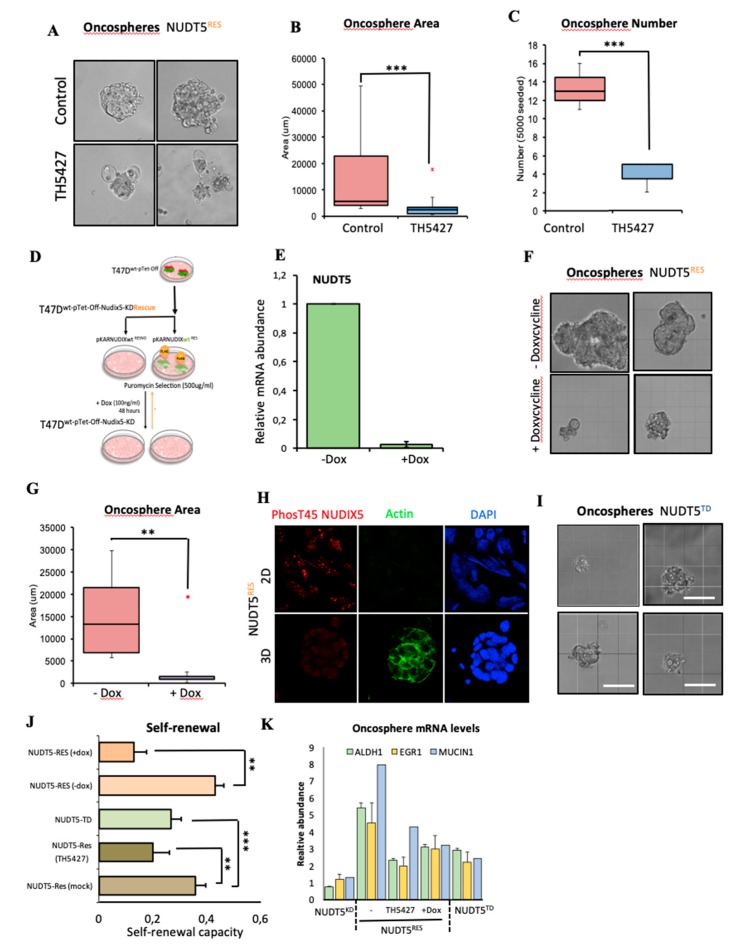
Oncosphere generation is dependent on the ATP-generating activities of NUDT5. (**A**) Bright field images of oncospheres formed following 5 days in culture in NUDT5^RES^ cell lines grown in the absence or presence of 1.5 nM TH5427. Scale bar represents 200 μM. (**B**) Quantification of the area of oncospheres formed in NUDT5^RES^ cell lines in the presence or absence of TH5427; mean ± SEM. (**C**) Quantification of the number of oncospheres formed in NUDT5^RES^ cell lines in the presence or absence of TH5427; mean ± SEM. (**D**) Stable knock-down and inducible pKAR rescue experimental overview in T47D^M^ cells. (**E**) Relative mRNA levels of NUDT5 following the addition of doxycycline to NUDT5^RES^ cell lines. (**F**) Bright field images of oncospheres formed following 5 days in culture in NUDT5^RES^ cell lines grown in the presence of doxycycline for 24 h after seeding. Scale bar represents 200 μM. (**G**) Quantification of the area of oncospheres formed in NUDT5^RES^ cell lines in the presence or absence of doxycycline, 24 h after seeding. (**H**) Immunofluorescent staining of phosphor-T45 NUDT5 in 2D and 3D cell culture conditions. (**I**) Bright field images of oncospheres formed following 5 days in culture in NUDT5^TD^ cell lines. (**J**) Quantification of second generation self-renewal capacity in NUDT5^RES^ (± TH5427), NUDT5^KD^ and NUDT5^TD^ cell lines, five days after primary 3D culture disassociation; data represents mean ± SEM. (**K**) qRT-PCR gene expression analysis of CSC marker; ALDH and EMT marker; EGFR1 and tumour marker; MUCIN1 in NUDT5^RES^ (±TH5427), NUDT5^KD^ and NUDT5^TD^ cell lines; data represents mean ± SEM.

**Figure 6 cancers-11-01337-f006:**
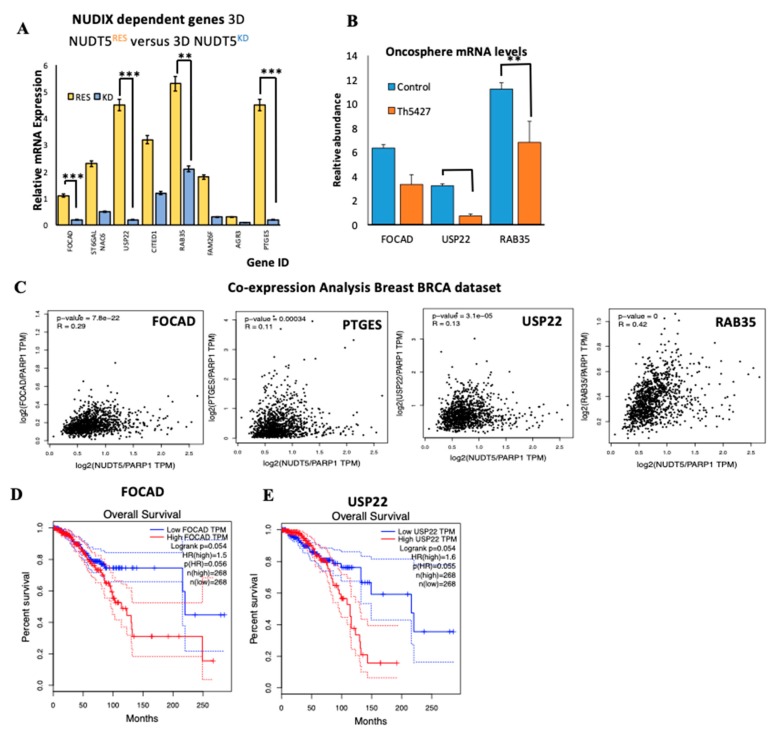
NUDT5 activity induces novel oncogenic drivers in the oncosphere associated with poor prognosis. (**A**) Relative mRNA expression levels of several genes in NUDT5^RES^ versus NUDT5^KD^ cell lines in 3D conditions. (**B**) Expression levels of focadhesin (*FOCAD*), ubiquitin specific peptidase 22 (*USP22*) and *RAB35* in NUDT5^RES^ oncospheres in the presence or absence of TH5427. (**C**) Co-expression of *FOCAD*, *USP22* and *RAB35* mRNA compared to *NUDT5* in TGCA breast tumour patient dataset (*n* = 2435). Kaplan Meyer patient data stratifying patients based on the expression levels of *FOCAD* (**D**) or *USP22* (**E**).

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
