# Peer review of "Expression of Oncogenic Drivers in 3D Cell Culture Depends on Nuclear ATP Synthesis by NUDT5"

_cancers, 2019, doi:10.3390/cancers11091337_

Round 1

Reviewer 1 Report

Katie Pickup et al investigate in their manuscript how the over-expression of NUDT5 in cancer cells contributes to the poorer prognosis of patients. They demonstrate that the suppression of NUDT5 or its enzymatic activity to generate ATP interferes with 3D oncosphere formation. They analyse the genes which are preferentially expressed in oncospheres when NUDT5 is present in comparison to oncospheres with suppressed NUDT5. Amongst these genes are known and novel markers which correlate with poor prognosis for cancer patients.  

The authors show that the suppression of NUDT5's enzymatic activity to generate ATP is sufficient to inhibit oncosphere formation. From this observation they conclude that it might be beneficial to focus drug development efforts on agents that suppress this activity of NUDT5.  

The manuscript is very rich in data guiding the reader well through the experiments. The conclusions are all sound and supported by the presented experimental evidence. I think the article can be published as it is. 

Attention has to be paid to the quality of the submitted figures. The text in the images did not make it properly through to the pdf file. Some letters are not printed, the formatting of some text blocks is erratic and their positioning in relation to the objects is sometimes wrong. 

Author Response

We thank the reviewer for their insightful comments and are appreciative of their interest in the paper and the results that are shown and that they feel that the article can be published as it is.

As they mentioned: "Attention has to be paid to the quality of the submitted figures. The text in the images did not make it properly through to the pdf file. Some letters are not printed, the formatting of some text blocks is erratic and their positioning in relation to the objects is sometimes wrong."

We apologise for this oversight and these formatting errors have now been corrected in the revised manuscript.

Reviewer 2 Report

The manuscript by Katie Pickup al. studies the roles of NUDT5 in breast cancer cell growth and sphere formation. The authors established tumorspheres from breast cancer cell line T47D as a breast oncosphere and examined the effects of NUDT5 expression on the growth and cloneginicity of these cells. The authors then examined the effects of NUDT5 inhibitor TH542 on ATP production during cancer sphere formation and demonstrated that this inhibitor can target breast cancer sphere in vitro. Overall the study focuses on an important topic and describes a potent regulator that preferentially targets breast cancer sphere. However, the manuscript will be improved if the authors address the following concerns.

In this study, authors have used T47D breast cancer cell line, which were derived from the pleural effusion of a ductal carcinoma found in the mammary gland of metastatic site. This cell line are distinct from other human breast cancer cells in that their progesterone receptors are not regulated by estrogen, a hormone that is abundant within the cells themselves. Authors need to show what is the scientific reason to use T47D cell line as an oncosphere moedel. Importantly, they need to prove NUDT5 function in cancer cell growth, metastasis, and sphere formation in other breast cell lines, such as MCF-7, MDA-MB-435, or HS-281. Authors only studied the roles of NUDT5 in breast cancer cell growth, migration, and sphere formation in vitro. Importantly, they need to confirm the role of NUDT5 in tumor formation and metastasis in vivo with xenograft animal model. It would be helpful to to demonstrate specificity of NUDT5 inhibitor TH542. Similarly, it is important to demonstrate that TH542 exerts selective effects on 2D cultured breast cells and 3D breast cancer sphere. Is the concentration of TH542 achievable in human physiological concentration? Clinical application should be addressed in discussion section.

Author Response

On behalf of all authors in this study I wish to thank the reviewer for their insightful and throughout review of our manuscript.

With this in mind we would like to take this opportunity to address each of the concerns/comments raised by the reviewer in turn with the hope that this will assure them that the manuscript it ready for publication in Cancers.

In this study, authors have used T47D breast cancer cell line, which were derived from the pleural effusion of a ductal carcinoma found in the mammary gland of metastatic site. This cell line are distinct from other human breast cancer cells in that their progesterone receptors are not regulated by estrogen, a hormone that is abundant within the cells themselves. Authors need to show what is the scientific reason to use T47D cell line as an oncosphere moedel. Importantly, they need to prove NUDT5 function in cancer cell growth, metastasis, and sphere formation in other breast cell lines, such as MCF-7, MDA-MB-435, or HS-281.

Response: NUDT5 was initially identified as a key component of progesterone-induced breast cancer cell growth in T47D cell lines (Wright et al 2016). The idea for the work presented here was based on the surprise result that the knockdown of NUDT5 had no effect on cells grown in culture under basal conditions (Figure 1) which is in stark contrast to the correlation with elevated NUDT5 levels and the prediction of a poorer outcome in cancer patient datasets (Figure S1) of different tissue of origin not only breast. Although the knockdown cell line used in the manuscript is T47D, with all respect we strongly believe that we have shown the dependence of NUDT5 and specifically ATP generation (CSC, sphere formation) in multiple different cancer cells lines within this manuscript including other breast cancer cell lines (Figure S1, S3). With leads to a more general conclusion regarding the role of NUDT5 in cancer in general.

Authors only studied the roles of NUDT5 in breast cancer cell growth, migration, and sphere formation in vitro. Importantly, they need to confirm the role of NUDT5 in tumor formation and metastasis in vivowith xenograft animal model.

Response: We fully agree with the reviewers comment regarding the importance of the analysis of NUDT5 in vivo. However, with respect we would like to stress that this kind of analysis is well known to be both financially and time consuming. We have ongoing lines of investigation regarding the development of a specific inhibitor of NUDT5 which only effects the ATP activity for clinical development. Such in vivo models are proposed for the future but we strongly believe are not required for the completion of this manuscript and the specific inhibitors and analysis of the in-vivo work will form the basis of a subsequent publication. In addition we feel that the work presented here is as it stands a complete body of work, which will be of great interest to the community at large.

It would be helpful to to demonstrate specificity of NUDT5 inhibitor TH542.

Response: We do not believe that the specificity of TH5427 is relevant for this manuscript as it has been thoroughly documented in a prior publication between our lab and the lab of Thomas Helleday in the Karolinska institute (Page et al. 2018 https://doi.org/10.1038/s41467-017-02293-7) However we do apologize if this was not made clear in the manuscript.

Similarly, it is important to demonstrate that TH542 exerts selective effects on 2D cultured breast cells and 3D breast cancer sphere.

Response: TH5427 has no effect on breast cancer cell growth under basal conditions however does inhibit progesterone induced (In T47D) or estrogen induced (in MCF7) proliferation as detailed in our prior publication (Page et al. 2018 https://doi.org/10.1038/s41467-017-02293-7). The data presented here in this publication aims to show that inhibition of NUDIX5 is a more general effect not only for the inhibition of breast cancer cells but other cancer cell types including colorectal, pancreatic, liver, renal and cervical cancer (Figure S1, S2 and S3) by effecting the specific oncogenic drivers and the cancer stem cell niche within, driving a more aggressive phenotype.

Is the concentration of TH542 achievable in human physiological concentration? Clinical application should be addressed in discussion section. 

Response: The analysis of TH5427 as mentioned previously has been characterized in a prior publication https://doi.org/10.1038/s41467-017-02293-7) In addition, as stated Th5427 inhibits both the AMP and ATP generating activities of NUDIX5 and therefore would not be suitable as a therapeutic compound based on side effects and the global effect of inhibiting the essential role of AMP synthesis within cells. The use of Th5427 within this publication forms part of a proof of concept; proving the mechanism (inhibiting CSC growth ) via the inhibition of NUDT5. Ongoing and future work is focused (as discussed in point 2), on the development of a more ATP specific inhibitor of NUDT5 for clinical development, which was stated within the conclusions section of the manuscript

patient datasets strongly indicates that any future drug discovery focusing on an ATP specific inhibitor of NUDT5 would be potentially beneficial for the treatment of cancers including breast and ovarian, reducing side effects, patient health and economic burden.”

Round 2

Reviewer 2 Report

All raised issues are mostly addressed.